# DNA Barcoding and Species Delimitation for Dogfish Sharks Belonging to the *Squalus* Genus (Squaliformes: Squalidae)

Ailton A. Ariza [1,*], Aisni M. C. L. Adachi [1], Pollyana Roque [2], Fabio H. V. Hazin [2,†], Marcelo Vianna [3], Matheus M. Rotundo [4], Sergio M. Delpiani [5,6], Juan M. Díaz de Astarloa [5,6], Gabriela Delpiani [5,6], Claudio Oliveira [1], Fausto Foresti [1] and Vanessa P. Cruz [1,*]

1 Laboratório de Biologia e Genética de Peixes, Instituto de Biociências de Botucatu, Universidade Estadual Paulista, Botucatu 18618-689, Brazil; aisnimayumi@gmail.com (A.M.C.L.A.); claudio.oliveira@unesp.br (C.O.); f.foresti@unesp.br (F.F.)

2 Laboratório de Oceanografia Pesqueira, Universidade Federal Rural de Pernambuco, Recife 52171-900, Brazil; pollyana_cgr@hotmail.com (P.R.); fhvhazin@terra.com.br (F.H.V.H.)

3 Laboratório de Biologia e Tecnologia Pesqueira, Universidade Federal do Rio de Janeiro, Rio de Janeiro 21941-901, Brazil; mvianna@biologia.ufrj.br

4 Acervo Zoológico da Universidade Santa Cecília, Universidade Santa Cecília, Santos 11045-907, Brazil; mmrotundo@unisanta.br

5 Grupo de Biotaxonomía Morfológica y Molecular de Peces, Departamento de Biología, Facultad de Ciencias Exactas y Naturales, Instituto de Investigaciones Marinas y Costeras, Funes 3250, Mar del Plata 7600, Argentina; matidelpiani16@yahoo.com.ar (S.M.D.); juanastarloa@gmail.com (J.M.D.d.A.); gabriela.delpiani@gmail.com (G.D.)

6 Consejo Nacional de Investigaciones Científicas y Técnicas, CABA AAJ, Buenos Aires 2290, Argentina

* Correspondence: ailton.ariza@unesp.br (A.A.A.); cruzvp@outlook.com (V.P.C.)

† In memorian.

**Abstract:** The *Squalus* genus comprises a group of small demersal sharks occurring circumglobally, popularly known as dogfish sharks. This genus exhibits a conserved morphology, thus making correct morphological identification difficult. Considering these taxonomic problems and the scarcity of molecular data, the present study aimed to identify *Squalus* genus MOTUs, using DNA barcoding for species delimitation via ABGD (automatic barcode gap discovery), PTP (Poisson tree process), and GMYC (general mixed Yule coalescent) employing the mitochondrial COI gene. A total of 69 sequences were generated from samples obtained from the American coast in both the Atlantic and Pacific Oceans. The ABGD analysis was the most conservative among the three applied delimitations, indicating three taxonomic units, while the PTP analysis revealed nine MOTUs, with two conflicting units noted between *S. clarkae* + *S. mitsukurii* and *S. albicaudus* + *S. cubensis*. The GMYC analysis indicated an excessive division, with *S. acanthias* and *S. mitsukurii* subdivided into six MOTUs each and *S. blainville*, into four. These findings demonstrated that *Squalus* presents a complex of previously defined species, with misidentified samples deposited in databases leading to difficulties in analyzing the real distribution and diversity of species belonging to this genus. Thus, further efforts to highlight possible new species are recommended.

**Keywords:** dogsharks; Elasmobranchii; mitochondrial DNA; COI; species identification

## 1. Introduction

Dogfish are cartilaginous fish belonging to the genus *Squalus* Linnaeus, 1758 (Squaliformes, Squalidae), comprising 35 described species [1–4]. Dogfish are small, migratory, demersal sharks living between 100 and 500 m in depth [5], reaching up to 1.5 m in length and presenting a wide global geographical distribution, occurring on continental shelves in the Atlantic, Pacific, and Indian oceans [5–7].

As with many other elasmobranchs, the life history of the *Squalus* genus is characterized by slow growth, late sexual maturation, long life expectancy, and low fecundity [8],

with individuals tending to aggregate by sex and size [6,9]. The dogfish is a yolk-sac viviparous species [10–12] with a long gestation period, estimated as lasting up to two years [13–15]. These attributes lead to low population growth rates and limited capacity to withstand fishing pressures, resulting in rapid population declines [16]. Because of their biology and anthropic actions, most *Squalus* species are currently classified as Threatened by the International Union for Conservation of Nature (IUCN) [17], whereas the Chico Mendes Institute for Biodiversity Conservation (ICMBIO), a Brazilian environmental agency linked to the Ministry of the Environment, categorizes only *S. acanthias* as Critically Endangered (CR) [18].

Overall, sharks belonging to the *Squalus* genus exhibit a conserved body morphology, making identification based solely on morphological characters problematic, leading to misidentifications [19]. This complexity is amplified even further by the high overlap of morphological characters among species, as identification is often based on limited and insufficiently consistent characters, such as number of vertebrae and morphometric data [2,4,5,20–22].

Reliable information on species richness is essential for any biodiversity study and conservation policies, although it is often difficult to discriminate a species based on highly similar morphological characters [20]. In this regard, reliable species identification is the first and most important step for the application of conservation policies and sustainable exploitation of natural resources [23], even more so considering the presently accelerated biodiversity crisis induced by human activities [24].

In recent years, different genetic studies have attempted to identify *Squalus* species using mitochondrial COI and NADH2 genes [3,21,25–28]. In general, three well-defined groups within the genus have been reported, namely group I, comprising *S. suckleyi* and *S. acanthias*; group II, comprising *S. blainville*/*S. megalops*/*S. raoulensis*/*S. brevirostris*; and group III, the *S. mitsukurii* complex, comprising *S. edmundsi*, *S. japonicus*, *S. grahami*, *S. clarkae*, and *S. mitsukurii* [21,22,26,29].

Generally, barcoding researchers have used a 2% divergence threshold as a heuristic cutoff value in fish species delimitation [30,31]. Nevertheless, it is already known that elasmobranchs have low evolutionary rates when compared with other fish species [32], which means that they are a more genetically conserved group [33–35]. Thus, in rays and sharks, we found some genera that presented about 1% of genetic distance between species, including among members of the genus of rays *Mobula* [36], the genus *Carcharhinus* [27,37,38], and the genus *Squalus* [3,21,25,26].

Considering the important taxonomic problems that characterize this group and the scarcity of available molecular data, the main goal of this study was to identify molecular operational taxonomic units (MOTUs) in the *Squalus* genus based on the analysis of sampled nominal species collected in different Western Atlantic and Pacific Ocean regions through the DNA barcoding technique employing the cytochrome c oxidase I (COI) genetic marker. The data were compared with available databases using species delimitation approaches including automatic barcode gap discovery (ABGD) [39], the Poisson tree process (PTP) [40], and the general mixed Yule coalescent (GMYC) [41,42], with the aim to add molecular data of the genus *Squalus* through the tools of species delimitation to assist in future works of integrative taxonomy.

## 2. Materials and Methods

### 2.1. Sample Collection

Samples were obtained from 69 dogfish shark specimens from the Western Atlantic and Pacific Oceans belonging to the *Squalus* genus (Squaliformes: Squalidae) representing three nominal species, *S. mitsukurii*, *S. albicaudus*, and *S. acanthias*, (Figure S1). The tissue samples were deposited at the LBGP ichthyological collection (Laboratório de Biologia e Genética de Peixes—Fish Biology and Genetics Laboratory) belonging to UNESP in Botucatu, Sao Paulo, Brazil. All samplings were performed in accordance with Brazilian

government standards (SISBIO protocol 13843-1) and an Animal Ethical Committee. Small muscle fragments (<1 cm$^2$) from each sample were obtained and preserved in 96% ethanol.

Total genomic DNA was isolated from muscle tissues of each specimen with a DNeasy Tissue Kit (Qiagen, Hilden, Germany) according to the manufacturer's instructions. Amplification reactions of mitochondrial gene cytochrome c oxidase subunit I (COI) were performed in a total volume of 12.5 μL, with 1.25 μL of 10× buffer (10 mM Tris-HCl + 15 mM MgCl$_2$); 0.5 μL dNTPs (200 nM of each); 0.5 μL each of the 5 mM primers L6252-Asn and H7271-COXI, as described in Melo et al. [43]; 0.2 μL of PHT Taq DNA polymerase (Phoneutria Biotecnologia e Serviços Ltd., Belo Horizonte, Brasil); 1 μL template DNA (12 ng); and 8.7 μL ddH$_2$O. The PCR reactions consisted of initial denaturation at 95 °C for 3 min; 25 cycles at 94 °C for 30 s, 52 °C for 45 s, and 68 °C for 1 min; and final extension at 68 °C for 7 min. All PCR products were first visually identified on a 1% agarose gel. The purified PCR products were sequenced using a Big Dye Terminator v3.1 Cycle. Sequencing was performed with the BigDye Terminator v3.1 Cycle Sequencing Kit (Applied Biosystems, Waltham, MA, USA). Individual reactions were performed with approximately 30 ng template PCR product, 3.2 pmol primer, 1 μL terminator mix, and 5 μL Better Buffer (The Gel Co., Eden Prairie, MN, USA) in a total volume of 15 μL. PCR sequencing profiles consisted of an initial denaturation step of 4 min at 96 °C followed by 30 cycles of 30 s at 96 °C, 15 s at 50 °C, and 4 min at 60 °C. Sequencing was carried out on an automated ABI 3130xl Applied Biosystems sequencer.

### 2.2. Barcoding

The COI sequences were edited in the Geneious 6.0 software(Biomatters, Ltd., Auckland, New Zeland) [44], with each sequence manually reviewed for uncalled and miscalled bases and all variable positions confirmed by comparing sequence reads produced by the forward and reverse sequences of each individual. A consensus sequence was produced for each individual, and all sequences were deposited in GenBank under accession numbers ON827418 to ON827486.

Aligned consensus sequences were compared with those deposited in the National Center for Biotechnology Information (NCBI) database (http://www.ncbi.nlm.nih.gov/, accessed on 27 March 2021) using the Basic Local Alignment Search Tool—Nucleotide (BLASTn). Each sequence used in this study is provided in a supplementary table (Table S1) and was later aligned using the Muscle algorithm [45] implemented within the Geneious 6.0 software [44]. All parameters followed the default version of the algorithm.

The sequences obtained herein were compared with 204 COI GenBank sequences [21,46–53] referring *to S. blainville, S. suckleyi, S. acanthias, S. brevirostris, S. clarkae, S. cubensis, S. japonicus, S. mitsukurii, S. grahami,* and *S. edmundsi*. The final dataset comprised 273 sequences from 11 *Squalus* species and 1 sequence for *Cirrhigaleus asper* (MN982926), representing an external group, totaling 274 sequences.

The best-fit model of nucleotide evolution for the data was estimated for the analyzed matrix with the MEGA X program [54], applying the neighbor-joining (NJ) method using the Kimura two-parameter (K2P) model [55]. Bootstrap replicates were assessed by applying 1000 replicates [56]. Trees were visualized and edited using the FigTree v1.4 program (www.tree.bio.ed.ac.uk/software/figtree, accessed on 27 March 2021), (Edinburgh, UK). The mean genetic inter- and intraspecific distances for nominal species were calculated under the K2P model and displayed in a pairwise distance matrix.

### 2.3. Automatic Species Delimitation Analyses

To infer *Squalus* species delimitation criteria based on a partial COI gene, molecular operational taxonomic unit (MOTU) estimations were performed by employing three molecular tools to delimit species. The first delimitation was conducted using automatic barcode gap discovery (ABGD) [39] run on the ABGD web server (https://bioinfo.mnhn.fr/abi/public/abgd/abgdweb.html, accessed on: 30 March 2021). All parameters followed

the default version of the program (model = Jukes–Cantor (JC69) Pmin = 0.001, Pmax = 0.1, steps = 10, X (relative gap width) = 1.5, number of bins = 20).

The second delimitation was performed applying the Poisson tree process (PTP) [40] based on a nonultrametric tree run on the PTP web server (https://species.h-its.org/ptp, accessed on 30 March 2021). The maximum likelihood (ML) tree was used as the input. The best model used in this dataset was selected based on AIC (i.e., had the lowest AIC) as estimated using the MEGA X software (https://www.megasoftware.net/citations, accessed on 27 March 2021) based on the best nucleotide substitution model HKY + G + I (5968.201). The PTP analysis was then performed for 100,000 generations MCMC, with a thinning value of 100 and burn-in of 0.1.

The third delimitation analysis was performed through the general mixed Yule coalescent (GMYC) method [41,42] run on the GMYC web server (https://species.h-its.org/gmyc/, accessed on 27 March 2021). The Elimdupes software (https://www.hiv.lanl.gov/content/sequence/ELIMDUPES/elimdupes.html, accessed on 27 March 2021) was used to group identical sequences and thus reduce the computational analysis time. The tree parameters were selected in the BEAUTI program belonging to the BEAST program package to calibrate the ultrametric tree, uncorrelated lognormal relaxed clock, and coalescence speciation models, where exponential growth was applied employing the HKY + G + I nucleotide substitution model. The MCMC method was performed for 10 million iterations. The Tracer v1.7 software was used to verify convergence (ESS > 200).

The Tree Annotator v1.8 software was used at a 10% burn-in, and the output file was submitted to the Figtree software to detect possible analysis errors such as polytomies or others. The output file was then submitted to the online GMYC version, applying the site's default parameters.

Genetic groups were selected based on a MOTU consensus, and mean genetic interspecific and intraspecific distances were calculated under the K2P model and displayed in a pairwise distance matrix.

To better understand the relationships among the three major *Squalus* groups known in the literature (group I, comprising *S. suckleyi* and *S. acanthias*; group II, comprising of *S. blainville; S. brevirostris, S. cubensis* and *S. albicaudus*; and group III, comprising *S. edmundsi, S. japonicus, S. grahami, S. clarkae* and *S. mitsukurii*), the number of variable sites, number of haplotypes, and haplotype diversity of each group were evaluated and estimated by the DnaSP v5 software [57] with the median-joining network produced by the PopArt program [58] for mutational analyses.

## 3. Results

A total of 69 sequences were generated from dogfish shark samples belonging to the *Squalus* genus, representing the nominal species *S. mitsukurii, S. albicaudus*, and *S. acanthias*. The amplification of the COI gene resulted in standardized 711 bp fragments, and the nucleotide composition analysis revealed a mean nucleotide composition of 24.8% adenine (A), 33.9% thymine (T), 16.5% guanine (G), and 24.7% cytosine (C). The dataset was submitted to the Basic Local Alignment Search Tool (BLAST) for correct identification by comparing the obtained results with sequences deposited in the NCBI database. Thirty *S. acanthias*, twenty-one *S. albicaudus*, and eighteen *S. mitsukurii* were identified, presenting 98.73 to 100% similarity. The *S. albicaudus* nomenclature was adopted for the sampled individuals, as they occurred on the Brazilian coast, considering that Viana et al. [2] described the occurrence of this species in the southeast Atlantic Ocean.

The matrix was complemented with 204 *Squalus* sequences obtained from GenBank [21,46–53] (Table S1), totaling 273 sequences for the final matrix analysis, representing 11 nominal species from different Mediterranean, Atlantic, and Pacific Ocean regions. The K2P distances of the COI sequence between species ranged from 0.72 to 8.3%, with the smallest and largest interspecific genetic distances identified between *S. albicaudus* and *S. cubensis* (0.0072) and *S. acanthias* and *S. brevirostris* (0.0832) (Table 1). Intraspecific genetic distances ranged from 0.0000 for *S. grahami* to 0.0043 for *S. mitsukurii*.

**Table 1.** Genetic distances (K2P) based on COI sequences among *Squalus* species (below the diagonal) and standard errors (above the diagonal). The numbers in bold represent the intraspecific K2P genetic distances.

| Species | 1 | 2 | 3 | 4 | 5 | 6 | 7 | 8 | 9 | 10 | 11 |
|---|---|---|---|---|---|---|---|---|---|---|---|
| 1—*S. suckleyi* | **0.0008** | 0.0031 | 0.0118 | 0.0119 | 0.0108 | 0.0109 | 0.0099 | 0.0105 | 0.0101 | 0.0095 | 0.0097 |
| 2—*S. acanthias* | 0.0077 | **0.0022** | 0.0117 | 0.0123 | 0.0110 | 0.0107 | 0.0099 | 0.0112 | 0.0107 | 0.0101 | 0.0101 |
| 3—*S. blainville* | 0.0785 | 0.0788 | **0.0032** | 0.0039 | 0.0050 | 0.0052 | 0.0105 | 0.0109 | 0.0104 | 0.0105 | 0.0106 |
| 4—*S. brevirostris* | 0.0792 | 0.0832 | 0.0116 | **0.0018** | 0.0053 | 0.0054 | 0.0106 | 0.0109 | 0.0106 | 0.0106 | 0.0107 |
| 5—*S. cubensis* | 0.0701 | 0.0748 | 0.0168 | 0.0188 | **0.0013** | 0.0026 | 0.0099 | 0.0103 | 0.0100 | 0.0096 | 0.0100 |
| 6—*S. albicaudus* | 0.0707 | 0.0729 | 0.0178 | 0.0197 | 0.0072 | **0.0016** | 0.0100 | 0.0102 | 0.0101 | 0.0102 | 0.0097 |
| 7—*S. edmundsi* | 0.0630 | 0.0636 | 0.0638 | 0.0667 | 0.0614 | 0.0618 | **0.0021** | 0.0054 | 0.0057 | 0.0047 | 0.0049 |
| 8—*S. japonicus* | 0.0646 | 0.0720 | 0.0655 | 0.0663 | 0.0601 | 0.0600 | 0.0190 | **0.0009** | 0.0051 | 0.0051 | 0.0055 |
| 9—*S. grahami* | 0.0631 | 0.0703 | 0.0634 | 0.0649 | 0.0603 | 0.0610 | 0.0200 | 0.0169 | **0.0000** | 0.0045 | 0.0048 |
| 10—*S. clarkae* | 0.0589 | 0.0675 | 0.0636 | 0.0659 | 0.0577 | 0.0661 | 0.0169 | 0.0185 | 0.0133 | **0.0029** | 0.0026 |
| 11—*S. mitsukurii* | 0.0605 | 0.0659 | 0.0644 | 0.0665 | 0.0638 | 0.0613 | 0.0190 | 0.0213 | 0.0154 | 0.0084 | **0.0043** |

A neighbor-joining tree based on COI gene sequencing identified three main clades (Figure 1), which represented a *suckleyi/acanthias* group (group I), comprising *S. suckleyi* and *S. acanthias*; the *S. blainville/S. megalops/S. raoulensis/S. brevirostris* group (group II), comprising *S. blainville*, *S. brevirostris*, *S. cubensis*, and *S. albicaudus*, and the *S. mitsukurii* complex group (group III), comprising *S. edmundsi*, *S. japonicus*, *S. grahami*, *S. clarkae*, and *S. mitsukurii*. These groups represented 11 nominal species, with bootstrap values ranging from 56 to 100%.

The species delimitation results indicated 3 MOTUs by the ABGD method, 9 by the PTP method, 24 by the GMYC method. Three taxonomic groups were identified based on the ABGD analysis, the most conservative among the three applied delimitation analyses, comprising *S. suckleyi* and *S. acanthias* (group I), *S. blainville*, *S. brevirostris*, *S. cubensis* and *S. albicaudus* (group II), and *S. edmundsi*, *S. japonicus*, *S. grahami*, *S. clarkae* and *S. mitsukurii* (group III).

The discrimination based on the PTP analysis identified nine MOTUs. *S. suckleyi*, *S. acanthias*, *S. blainville*, *S. brevirostris*, *S. edmundsi*, *S. japonicus*, and *S. grahami* formed one group, with the other two MOTUs formed were by the pairs of nominal species *S. cubensis* and *S. albicaudus* and *S. clarkae* and *S. mitsukurii*.

The GMYC analysis detected a larger number of groups, totaling 24 MOTUs. A single MOTU grouped *S. suckleyi*, *S. brevirostris*, *S. cubensis*, *S. albicaudus*, *S. edmundsi*, *S. japonicus*, *S. grahami*, and *S. clarkae*. The highest number of MOTUs was noted mainly for *S. acanthias* individuals, at six, while *S. blainville* individuals presented four MOTUs, and *S. mitsukurii* individuals, six (see Table S1).

An investigation of pairwise genetic distances based on the PTP analysis in nine MOTUs was carried out, resulting in interspecific genetic variation values of 0.0032 between *S. blainville* and *S. brevirostris* and of 0.0832 between *S. acanthias* and *S. brevirostris* (Table 2). Intraspecific genetic distance values were 0.0000 for *S. grahami* and 0.0055 for *S. mitsukurii*.

The median-joining network was also used for each of the three major *suckleyi/acanthias* lineages forming the *S. blainville/S. megalops/S. raoulensis/S. brevirostris* and *S. mitsukurii* complex groups. This analysis has the advantage of showing the history (step by step) of the mutations connecting nodes between samples or species. A total of 23 haplotypes were identified in the *suckleyi/acanthias* group, with a haplotype diversity of 0.6962 and 31 variable sites. Among the 82 analyzed sequences, the haplotypes were subdivided into two main groups, one composed by S. *suckleyi* haplotypes and the other by *S. acanthias* haplotypes (Figure 2A).

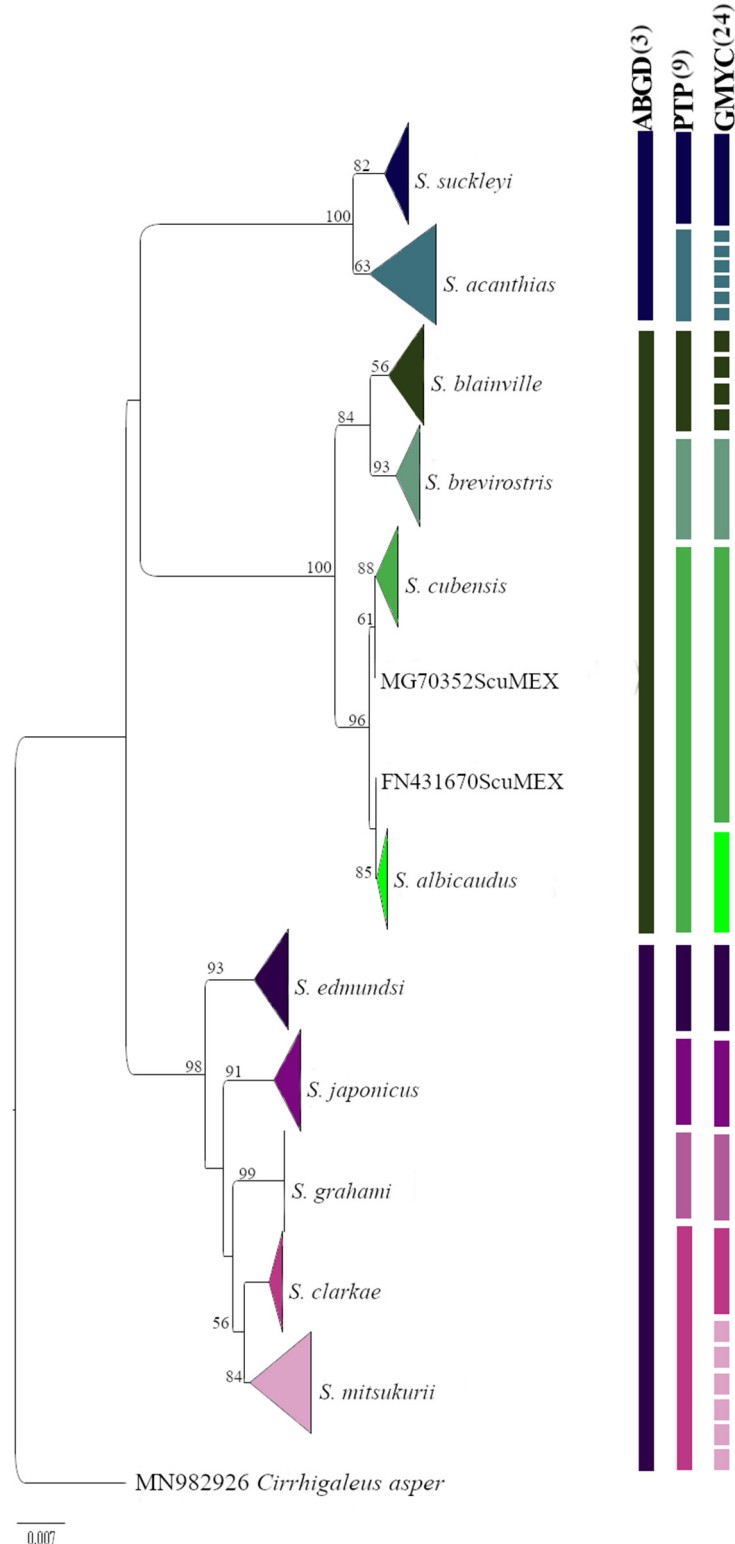

**Figure 1.** Neighbor-joining tree based on the COI gene from 11 nominal *Squalus* species with bootstrap values on branches. On the right, the vertical bars represent the division into MOTUs (molecular operational taxonomic units) obtained by ABGD (automatic barcode gap discovery for primary species delimitation), PTP (Poisson tree process), and GMYC (generalized mixed Yule coalescent) analyses.

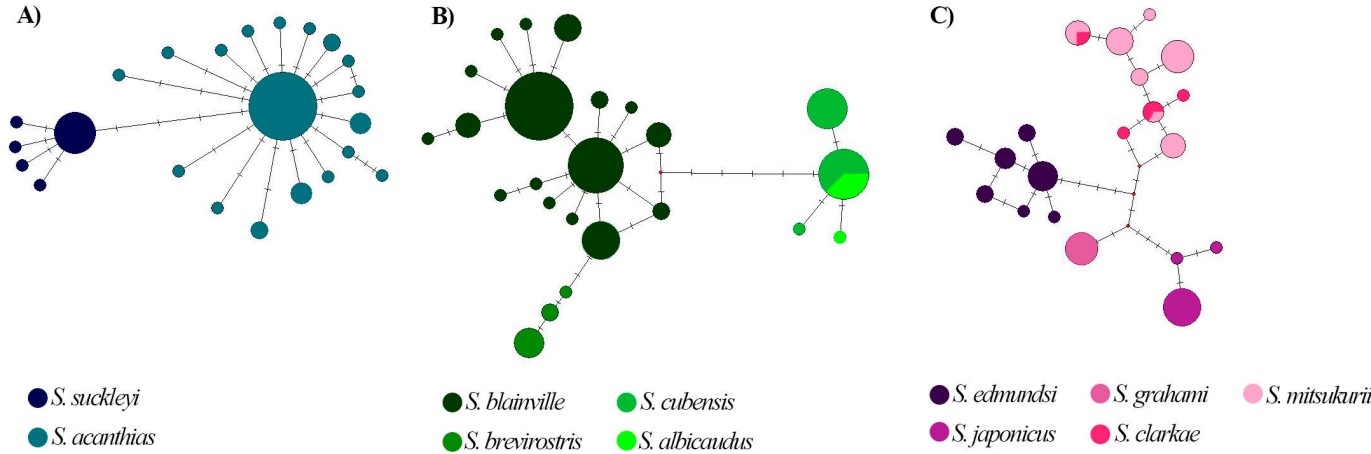

**Figure 2.** Median-joining network of the three identified groups based on COI gene sequencing from 11 nominal *Squalus* species: (**A**) group I, comprising *S. suckleyi* and *S. acanthias*; (**B**) group II, consisting of *S. blainville*; *S. brevirostris*, *S. cubensis*, and *S. albicaudus*; (**C**) group III, grouping *S. edmundsi*, *S. japonicus*, *S. grahami*, *S. clarkae*, and *S. mitsukurii*. The dashes represent mutational steps. The size of the circle representing each haplotype is proportional to the number of individuals within that haplotype.

**Table 2.** Genetic distances (K2P) based on the division of MOTUs obtained through a PTP analysis among *Squalus* species (below the diagonal) and standard errors (above the diagonal). The numbers in bold represent the intraspecific K2P genetic distances.

| Species | 1 | 2 | 3 | 4 | 5 | 6 | 7 | 8 | 9 |
|---|---|---|---|---|---|---|---|---|---|
| 1—*S. suckleyi* | **0.0008** | 0.0031 | 0.0121 | 0.0119 | 0.0108 | 0.0106 | 0.0108 | 0.0104 | 0.0100 |
| 2—*S. acanthias* | 0.0078 | **0.0022** | 0.0121 | 0.0122 | 0.0107 | 0.0104 | 0.0114 | 0.0110 | 0.0103 |
| 3—*S. blainville* | 0.0785 | 0.0788 | **0.0038** | 0.0041 | 0.0052 | 0.0107 | 0.0112 | 0.0107 | 0.0106 |
| 4—*S. brevirostris* | 0.0792 | 0.0832 | 0.0032 | **0.0018** | 0.0056 | 0.0110 | 0.0112 | 0.0108 | 0.0107 |
| 5—*S. cubensis +S. albicaudus* | 0.0705 | 0.0734 | 0.0176 | 0.0194 | **0.0038** | 0.0100 | 0.0101 | 0.0100 | 0.0096 |
| 6—*S. edmundsi* | 0.0630 | 0.0636 | 0.0638 | 0.0667 | 0.0617 | **0.0021** | 0.0053 | 0.0055 | 0.0050 |
| 7—*S. japonicus* | 0.0646 | 0.0720 | 0.0655 | 0.0663 | 0.0600 | 0.0190 | **0.0009** | 0.0051 | 0.0055 |
| 8—*S. grahami* | 0.0631 | 0.0703 | 0.0634 | 0.0649 | 0.0608 | 0.0200 | 0.0169 | **0.0000** | 0.0045 |
| 9—*S. clarkae + S. mitsukurii* | 0.0602 | 0.0662 | 0.0642 | 0.0664 | 0.0623 | 0.0186 | 0.0208 | 0.0151 | **0.0055** |

Among *blainville/brevirostris* group individuals, haplotypes were categorized into three main groups among the 128 analyzed sequences. The three groups were composed by *S. blainville*, *S. brevirostris*, and *S. cubensis* and *S. albicaudus*, the latter two of which shared a total of 24 haplotypes (Figure 2B), with a haplotype diversity of 0.8674 and 25 variable sites. The median-joining network for the *S. mitsukurii* complex group, with 63 analyzed sequences, indicated specific haplotypes for *S. edmundsi*, *S. japonicus* and *S. grahami*, whereas *S. clarkae* and *S. mitsukurii* shared haplotypes among species (Figure 2C), totaling 20 haplotypes, with a haplotype diversity of 0.9360 and 25 variable sites.

## 4. Discussion

This study provides a wide genetic analysis aiming at identifying molecular operational taxonomic units (MOTUs) for *Squalus* specimens from the Western Atlantic Ocean. The data allowed for the identification of at least nine *Squalus* lineages among the 11 nominal analyzed species, comprising *S. suckleyi*, *S. acanthias*, *S. blainville*, *S. brevirostris*, *S. edmundsi*, *S. japonicus*, *S. grahami*, *S. cubensis* + *S. albicaudus*, and *S. clarkae* + *S. mitsukurii*.

Genetic distance employing DNA barcoding is a strong indicator of lineages or species Ward et al. [59], Hubert et al. [60], and Pereira et al. [61] suggested COI distances from

1% [60] to 2% [59,61] as thresholds for fish species. However, as highlighted by Ramirez et al. [62], such values were derived from comparative analyses among phylogenetically diverse groups, whereas DNA barcoding analyses of closely related groups of species may result in lower values [21,25,61,63,64].

Our results coupled to dogfish DNA barcode genetic distances indicated that, among the 54 analyzed comparative values, 41, representing 74.6% of the total data, were higher than 2%, reaching up to 8%. Meanwhile, 12 estimates, representing 21.8% of the total data, exhibited values around 1%. Values were lower than 1% in only 2 estimates, representing 3.6% of the analyzed species. Ziadi-Künzli et al. [25] found similar proportions to those detected herein in an analysis of 27 *Squalus* groups/lineages employing the COI gene, identifying that 66.4% of the estimates displayed genetic distances greater than 2%, while 26% of the estimates were around 1%, and 7.6% of the estimates were below 1%.

An alternative for carrying out species delimitation, especially in cases in which genetic distance values are below 1%, is the application of multiple "automatic species delimitation" methods, which provides an efficient approach in identifying putative species, or MOTUs [65]. The ABGD species delimitation analysis is known to result in conservative delimitation values and be unlikely to partition variations into species [66]. This method identified only three major *Squalus* strains, namely *S. suckleyi* and *S. acanthias* (group I), *S. blainville*, *S. brevirostris*, *S. cubensis*, and *S. albicaudus* (group II), and *S. japonicus*, *S. edmundsi*, *S. grahami*, *S. clarkae*, and *S. mitsukurii* (group III). Verissimo et al. [21] investigated *Squalus* species by employing the COI and NADH2 mitochondrial genes in 19 nominal species and also detected three major lineages. Thus, the present results were in accordance with previous studies [22,25,26].

Formerly, group I was represented only by *S. acanthias*. However, Ebert et al. [9] resurrected S. *suckleyi*, an endemic species from the North Pacific, and allocated it within this group. Our results supported the separation of *S. acanthias* and *S. suckleyi*, despite a genetic distance of below 1%. We identified one MOTU for *S. acanthias* and one MOTU for *S. suckleyi* according to the respective nominal species by means of the PTP analysis. The GMYC analysis also detected one MOTU for *S. suckleyi* but identified six MOTUs for *S. acanthias*. Ebert et al. [9] highlighted that numerous synonyms for *S. acanthias* are in place, with regional subspecies within this subgroup for the North Atlantic Ocean, the Black Sea, and the west coast of Southern Africa. A relationship among the analyzed locations was not, however, detected in the present work. Interestingly, the haplotype network revealed that these species did not share haplotypes.

The four nominal species that were part of group II, *S. blainville*, *S. brevirostris*, *S. cubensis* and *S. albicaudus*, exhibited high COI distance values (>1.9%, Table 1), but group II presented the lowest interspecific genetic distance detected herein, that between *S. cubensis* and *S. albicaudus* (0.72%). PTP analyses indicated one MOTU for *S. blainville*, one for *S. brevirostris*, and only one for *S. cubensis* + *S. albicaudus*. The GMYC analysis identified seven MOTUs, four for *S. blainville*, one for *S. blainville*, one for *S. cubensis*, and one for *S. albicaudus*. Other authors have indicated the existence of more than one species, requiring a taxonomic revision [26], and highlighted the diversity among individuals identified as *S. blainville*. In this context, it is also interesting to note the close relationship with *S. brevirostris* observed in the haplotype network, with only two mutations of difference.

Low levels of genetic variation among species of elasmobranchs with the COI gene have been reported in the literature [21,27,37,38,67]. These were also detected in this study among some members of the genus *Squalus*. The low rates found may have been due to the evolutionary aspects of the group, as it is already known that sharks and rays have lower evolutionary rates than other fish species [32], or even a recent speciation process.

In a taxonomic review of the *Squalus* genus occurring in the Southeastern Atlantic Ocean performed by Viana et al. [2], the authors diagnosed individuals previously identified as *S. cubensis* in the region, with a new species described as *S. albicaudus*. In the present study, the only delimitation analysis able to separate these two nominal species was the GMYC, although the haplotype network analysis revealed haplotype sharing between

them. One hypothesis is that *S. albicaudus* may comprise an *S. cubensis* population in the Southeastern Atlantic Ocean currently undergoing a speciation process, which may still be very recent and incomplete, as these species still share haplotypes. As mentioned by other authors, elasmobranch speciation is very common [67,68] and boundaries among populations or species are often difficult to detect.

The complexity detected in group III herein was noted in *S. mitsukurii* and *S. clarkae*, which exhibited low COI distance values (1%); distances were greater than 1.3% for the other species (Table 1). Pfleger et al. [4] reported a 2.8% divergence between *S. clarkae* in the Gulf of Mexico and *S. mitsukurii* in Japan when employing the COI technique [27].

Herein, the GMYC method identified similar results to those of the PTP analysis but divided *S. clarkae* into a single MOTU and subdivided *S. mitsukurii* into six MOTUs. The haplotype network revealed that *S. japonicus*, *S. edmundsi*, and *S. grahami* did not share haplotypes, unlike *S. clarkae* and *S. mitsukurii*, which did. *Squalus mitsukurii* was originally described in Japan by Jordan and Snyder (1903), and despite identification issues due to morphological character overlapping, this species presents a circumglobal distribution [69], and its occurrence has likely been overestimated [4,70,71].

## 5. Conclusions

Our approach, using molecular tools for species delimitation, presented data to assist in future studies of species delimitation in the genus *Squalus*, since in many cases morphological data by themselves are not decisive. However, molecular data alone do not replace traditional taxonomy in the delimitation of species [72]. This integrative approach has been used over the years and has proven to be quite effective in elasmobranchs [72–76] and in other groups of organisms [76,77].

It is important to emphasize that the use of MOTUs represents an initial approach to support specific integrative analyses aiming for the identification of taxonomic groups [65]. However, because of the difficulty of morphologically defining *Squalus* species, many sequences available in genetic databases, i.e., BOLD and GenBank, indicate misidentifications or identifications only at the genus or family levels, making them not very useful for molecular identification purposes. Incorrect identifications or identifications at a higher taxonomic level often reflect high numbers of BINs, which are generally associated with ghost species but may also indicate undescribed species [5,75,78–80]. We also highlight that the barcode DNA in fish often does not reveal the genetic peculiarities existing in the groups, mainly in species with taxonomic complexity such as that already known to exist in *Squalus* [3,21,25,26], resulting in the need to use other genetic markers [36,81–83], or associations with morphological studies, for an integrated taxonomic approach [73,74,83].

**Supplementary Materials:** The following supporting information can be downloaded at: https: //www.mdpi.com/article/10.3390/d14070544/s1, Figure S1: Samples collected from individuals of the genus *Squalus* in the Western Atlantic Ocean and Pacific Ocean; Table S1. Species, locality, Sequence ID, analyses in this study, and GenBank accession numbers of specimens of *Squalus*.

**Author Contributions:** Conceptualization, F.H.V.H., M.M.R., J.M.D.d.A., G.D., C.O., F.F. and V.P.C.; methodology, A.A.A., A.M.C.L.A. and M.M.R.; validation, A.A.A.; formal analysis, A.A.A., M.V. and V.P.C.; investigation, A.A.A., S.M.D., J.M.D.d.A., G.D. and F.F.; resources, C.O.; data curation, P.R., M.V., S.M.D. and G.D.; writing—original draft preparation, A.A.A., S.M.D., J.M.D.d.A., A.M.C.L.A., C.O., F.F. and V.P.C.; writing—review and editing, A.A,.A., P.R., M.V., M.M.R., C.O., F.F. and V.P.C.; supervision, F.H.V.H.; project administration V.P.C.; funding acquisition, F.F. All authors have read and agreed to the published version of the manuscript.

**Funding:** This research was funded by the Conselho Nacional de Desenvolvimento Científico e Tecnológico (CNPq, grants to V.P.C., 107761/2019-0 to A.A.A.). C.O. received financial support from Fundação de Amparo à Pesquisa do Estado de Sao Paulo—FAPESP grants 2018/20610-1, 2016/09204-6, and 2014/26508-3, and from Conselho Nacional de Desenvolvimento Científico e Tecnológico—CNPq proc. 306054/2006-0 (C.O.).

**Institutional Review Board Statement:** All samples were collected in strict accordance with the regulations of the Brazilian Federal Animal Ethics Committee (SISBIO 13843–1), and the analyses followed the International Guidelines for Animal Experiments, as authorized by CEEAA IBB/UNESP, protocol number 556.

**Informed Consent Statement:** Not applicable.

**Data Availability Statement:** The data presented in the present study will be deposited and made available openly through the GenBank genetic sequence database.

**Acknowledgments:** We would like to thank the UNESP Biosciences Institute for the infrastructure provided,, Fundação de Amparo à Pesquisa do Estado de Sao Paulo—FAPESP and from Conselho Nacional de Desenvolvimento Científico e Tecnológico—CNPq We are grateful for the samples provided by Gavin Naylor, Florida Museum of Natural History, University of Florida, Gainesville, FL, USA.

**Conflicts of Interest:** The authors declare no conflict of interest.

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
