# Peer review of "DNA Barcoding and Species Delimitation for Dogfish Sharks Belonging to the Squalus Genus (Squaliformes: Squalidae)"

_diversity, doi:10.3390/d14070544_

Round 1
Reviewer 1 Report
Summary
The authors of this research article aim to clarify and identify species hypotheses in the genus Squalus, whom current recognized species present a high degree of morphological conservatism. The authors sequenced a single mitochondrial locus of species present along the coasts of the American continent and soundly applied an integrative taxonomical approach : identify species hypotheses based on genetic data with different methods and compare them to prior species hypotheses based on morphological data (i.e., currently recognized Squalus species).
They uncovered extensive cryptic diversity, with its potential for species new to science, while supporting recent works on the genus taxonomy. This publication will have important implication for species conservation and management, as it gives some guidelines to identify species hypotheses in the genus. Integrative taxonomy and species delineation methods based on molecular data are still rarely used, especially in in elasmobranch, and this publication shows the importance of questioning species hypotheses based on morphology solely.
General comments:
The manuscript is sometimes a little messy, especially in the results/discussion section, where understanding what is a result and what is in the literature can be tricky (regarding genetic distance between clades especially). The message needs to be clearer so we can properly assess what is new and what has been done.
The authors properly used part of an approach called Integrative taxonomy but never named it. This needs to be corrected. The scientific reasoning and protocols in the manuscript are well sounded, even though sometimes they need more explanations and details so they can be replicated. One analysis seems unnecessary to me right now, and the rationale behind it needs to be explained in regards of what it adds when compared to other species hypotheses delineation.
One main issue is the absence of the molecular lab section in the Material and Methods. Similarly, the section regarding the reconstruction of phylogenetic trees needs to be reworked, completed, and corrected.
Additionally, a general conclusion is missing to underline the key findings of this study and the next steps.
Introduction:
Line 44-45: Why are you giving information about Brazil? This is global, stay global. Local information is superfluous.
Line 49 – 50: Why only spiny dogfhish? rephrase if it is the only species with known information.
Line 64-66 : What kind of analyses are you referring to? I find this sentence quite confusing.
Line 71-76: you stated line 40-41 that there are 3 species in the Squalus genus, but here there are 11 names. Where does the difference come from? How many currently recognized species is there, based on which characters?
Line 77: How is it scarce? You do not provide any way for the reader to assess this, especially since you state in the previous paragraph that several studies worked on this subject before.
Line 83-84: Even though the approach is correct, there seems to be a misconception. These molecular species delineation (or delimitation, both terms are correct I believe) are not designed to identify species per se but species hypotheses. These hypotheses are then compared with other types of data (morphology, distribution, ecology, etc.) to properly delineate species and list their identification characters. This is called integrative taxonomy, and the term needs to appear somewhere in the manuscript, especially since this approach is not often used in this taxon (the elasmobranchs).
Materials and Methods
Line 89 : could you provide the detail of the sampling scheme? In other words, at least the basic metadata: which species, where (precisely if possible) and when? The figure does not provide this information.
Figure S1: Squalus needs to be italicized.
Line 96: how did you get the muscle samples? The animals were sacrificed? Or do you take a biopsy sample in the field?
Line 106: I suppose you used the default parameters for Muscle alignment? State it.
Line 109: Include in Table S1 all sequences and accession number. Add a column to the table giving the origin of the sequence (this study or otherwise). It is necessary to be able to reproduce your results.
Line 114: Are there several alignments (I suppose that’s what you imply by matrix)? Why? There is only one locus, so why would there be several matrices?
Line 114-115: You should use JModeltest2 to assess the most likely evolutionary model, and only then reconstruct your phylogenetic tree. Additionally, even if NJ trees often have the same topology as the real tree when the signal is strong, you should only present the ML tree topology, and add on each node its support with the ML and Bayesian reconstruction methods (see Erixon et al. 2003 Systematic Biology for the rationale behind it). Bayesian trees can be made with BEAST, or the MrBayes plug-in Geneious if I recall.
Line 118-119: correct the distance matrix calculation method according to my previous comment. If you choose K2-P because of low genetic distances, state it and refer to Hebert et al. (2003) Biological identifications through DNA barcodes. Proc R Soc Lond B Biol Sci, 270.
Line 127: why did you change the Pmax parameter?
Line 130: This should be move upward, and why did you use this evolution model if you did not test for the most likely model?
Line 136-140: this sentence is way too long and unstructured. Tree reconstruction should be put together and separated from species delineation methods: one uses the output of the others. Again, why did you use this evolution model if you did not test for the most likely model?
Line 147: what do you mean by motu consensus? And what are genetic groups? Species hypotheses?
Line 150: Are you talking about the groups you identified, or the three ones used in the literature (in the introduction)? In one case it is a different way to group individuals to identify species hypotheses based on haplotype distance, without an underlying evolutionary model, so close to the core idea of ABGD. In the other, you are representing sub-parts of your phylogenetic tree without the underlying evolutionary model to calculate distances, which seems incorrect. Please develop the rationale or delete from the manuscript.
Results
Line 163: what is the “S. albicaudus nomenclature”? Do you mean that this species name is not validated? Please explain.
Figure1: Please add the support value of each node and refer to my previous comment about phylogenetic tree reconstruction.
Line 167: Again, you state in the introduction “the Squalus genus Linnaeus, 1758 (Squaliformes, Squalidae), comprising three described species”, and here you talk about 11 species. Which should be named morphospecies as they are recognized solely on morphological data (that is the whole issue here, right?).
Line 197-199: This method is looking for a barcoding gap: if the genetic distance is extremely low between species having recently diverged but relatively higher between groups of closely related species, it may group independent “real” species into species hypotheses. This could be the case here.
Line 205-209: This method is known to overestimate the number of species (Zhang et al. 2013, Bioinformatics), this is not really a surprise, especially since it aims to identify independent lineages: if there is a strong population structure, GMYC might identify genetically different populations as species hypotheses. Could it be the case with the dogfish, based on its ecology?
Line 221-234: These haplotype networks are a different representation of the leaves of the phylogenetic tree. Please explain what this figure and analysis add to the other results.
Discussion
Line 258 - 264: I am quite confused with this paragraph. What are you saying?,What is the message? Also, some of this seems to be results?
Line 266 and on: see my previous comment about the ABGD-GMYC methods and enrich the discussion. Beef up the discussion about these methods and include Integrative taxonomy references.
Line 288: This sentence seems to be contradicting itself? Species showing high genetic distance but also the lowest?
Line 327: A general conclusion is missing here in my opinion? What is the main message and what will be the next step. Phylogenetic trees, especially of a single marker, do not always reflect species evolution and phylogenies must be discussed in an integrative framework that includes all available information (Padial, J.M., Miralles, A., la Riva De, I., Vences, M., 2010. The integrative future of taxonomy. Front. Zool. 7, 16.)
Author Response
Botucatu (Brazil), June 25, 2022.
Dear Ms. Tina Fu
Assistant editor
Please find enclosed the manuscript "DNA barcoding and species delimitation for
dogfish sharks belonging to the Squalus genus (Squaliformes: Squalidae)" by Ariza
et al. to be considered for publication in Diversity.
This manuscript is a revised version of our previous submitted manuscript with the
number ID 1772734. Following your request, we have detailed our response for the
reviewer’s in the following pages. In order to provide clarity, we have included the
reviewer’s comments in italics and our answer just below in bold.
We believe that reviewers’ comments have been adequately addressed through our
responses. We thank you for your time towards improving this manuscript, which we
hope is now acceptable for publication in Diversity.
Ailton A. Ariza.
Reviewer 1
Introduction:
Line 44-45: Why are you giving information about Brazil? This is global, stay global. Local
information is superfluous.
We agree with the reviewer and removed the sentence that mentions the local issue of
species of the genus.
Line 49 – 50: Why only spiny dogfhish? rephrase if it is the only species with known information.
In fact, the nomenclature used in the sentence was referring only to one species, but
the information is characteristic of the genus, so we corrected the sentence.
Line 64-66 : What kind of analyses are you referring to? I find this sentence quite confusing.
We revised the text and correct this sentence.
Line 71-76: you stated line 40-41 that there are 3 species in the Squalus genus, but here there are
11 names. Where does the difference come from? How many currently recognized species
is there, based on which characters?
We revised the text and correct this sentence. In the literature presented there are 35
valid species, there was a typing error.
Line 77: How is it scarce? You do not provide any way for the reader to assess this, especially
since you state in the previous paragraph that several studies worked on this subject before.
We revised the text and correct this sentence.
Line 83-84: Even though the approach is correct, there seems to be a misconception. These
molecular species delineation (or delimitation, both terms are correct I believe) are not
designed to identify species per se but species hypotheses. These hypotheses are then
compared with other types of data (morphology, distribution, ecology, etc.) to properly
delineate species and list their identification characters. This is called integrative
taxonomy, and the term needs to appear somewhere in the manuscript, especially since this
approach is not often used in this taxon (the elasmobranchs).
We agree with the note made by the reviewer and introduce a sentence exemplifying
this note.
Materials and Methods
Line 89: could you provide the detail of the sampling scheme? In other words, at least the basic
metadata: which species, where (precisely if possible) and when? The figure does not
provide this information.
Data such as location and number of species collected are shown in Table S1, but we
added to Figure S1 the region where the individuals were sampled.
Figure S1: Squalus needs to be italicized.
We checked the format and corrected the typo.
Line 96: how did you get the muscle samples? The animals were sacrificed? Or do you take a
biopsy sample in the field?
These animals were caught in fisheries, and our collaborators remove a small
fragment of the dead animals.
Line 106: I suppose you used the default parameters for Muscle alignment? State it.
We used the Muscle alignment default parameters, a sentence evidencing this was
added to the text.
Line 109: Include in Table S1 all sequences and accession number. Add a column to the table
giving the origin of the sequence (this study or otherwise). It is necessary to be able to
reproduce your results.
The sequences were sent to genbank, and the access codes were entered in
supplementary table S1.
Line 114: Are there several alignments (I suppose that’s what you imply by matrix)? Why? There
is only one locus, so why would there be several matrices? ok
In fact there is only one matrix, let's consider the comment and rewrite the sentence.
Line 114-115: You should use JModeltest2 to assess the most likely evolutionary model, and only
then reconstruct your phylogenetic tree. Additionally, even if NJ trees often have the same
topology as the real tree when the signal is strong, you should only present the ML tree
topology, and add on each node its support with the ML and Bayesian reconstruction
methods (see Erixon et al. 2003 Systematic Biology for the rationale behind it). Bayesian
trees can be made with BEAST, or the MrBayes plug-in Geneious if I recall.
We objective of this study was to carry out analyzes of delimitations of species in the
genus Squalus, for this reason we performed only the NJ and ML tree (the best model
was used the best nucleotide substitution model GTR+G+I) to better elucidate the
results obtained by the delimitations, and for that reason, we did not perform
phylogenetics Bayesian in the group.
Line 118-119: correct the distance matrix calculation method according to my previous comment.
If you choose K2-P because of low genetic distances, state it and refer to Hebert et al.
(2003) Biological identifications through DNA barcodes. Proc R Soc Lond B Biol Sci, 270.
We used the K2P genetic distance method to compare with other studies (which also
used the same parameter) with greater reliability.
Line 127: why did you change the Pmax parameter?
In fact, we use the default (changed in the text).
Line 130: This should be move upward, and why did you use this evolution model if you did not
test for the most likely model?
We use the most likely model according to the AIC method.
Line 136-140: this sentence is way too long and unstructured. Tree reconstruction should be put
together and separated from species delineation methods: one uses the output of the others.
Again, why did you use this evolution model if you did not test for the most likely model?
We revised the text, however, this sentence explains how we developed the input tree
for the GMYC analysis, again, we used the best models for the dataset obtained in this
study.
Line 147: what do you mean by motu consensus? And what are genetic groups? Species
hypotheses?
In this study, according to based on the division of MOTUs obtained in PTP analysis,
we analyze the mean genetic inter-specific and intraspecific distances were calculated
under the K2P model .
Line 150: Are you talking about the groups you identified, or the three ones used in the literature
(in the introduction)? In one case it is a different way to group individuals to identify
species hypotheses based on haplotype distance, without an underlying evolutionary
model, so close to the core idea of ABGD. In the other, you are representing sub-parts of
your phylogenetic tree without the underlying evolutionary model to calculate distances,
which seems incorrect. Please develop the rationale or delete from the manuscript.
We analyzed three large groups well known in the literature. In addition, the results
obtained here according to ABGD, formed these three large groups as explained in
Results and Discussion: Group I, comprising S. suckleyi and S. acanthias; Group II,
consisting of S. blainville; S. brevirostris, S. cubensis and S. albicaudus; Group III,
grouping S. edmundsi, S. japonicus, S. grahami, S. clarkae and S. mitsukurii. We will
not exclude, because this analysis shows mutations between and within the species of
each group, therefore, important for better elucidation.
Results
Line 163: what is the “S. albicaudus nomenclature”? Do you mean that this species name is not
validated? Please explain.
The nomenclature S. albicaudus is a valid species, it is an endemic species from the
Southeast of the Atlantic, described by Viana in 2016 [2] which was previously
believed to be a population of S. cubensis occurring in the Southeast of the Atlantic.
Figure1: Please add the support value of each node and refer to my previous comment about
phylogenetic tree reconstruction.
We considered the comment and added to the tree the support vaule of each node.
Line 167: Again, you state in the introduction “the Squalus genus Linnaeus, 1758 (Squaliformes,
Squalidae), comprising three described species”, and here you talk about 11 species.
Which should be named morphospecies as they are recognized solely on morphological
data (that is the whole issue here, right?).
We revised the text and correct this sentence. In the literature presented there are 35
valid species, there was a typing error.
Line 205-209: This method is known to overestimate the number of species (Zhang et al. 2013,
Bioinformatics), this is not really a surprise, especially since it aims to identify independent
lineages: if there is a strong population structure, GMYC might identify genetically
different populations as species hypotheses. Could it be the case with the dogfish, based on
its ecology?
Yes, it could be populations, probably related to their distribution.
Line 221-234: These haplotype networks are a different representation of the leaves of the
phylogenetic tree. Please explain what this figure and analysis add to the other results.
We revised the text and add explain this analysis in the text.
Discussion
Line 258 - 264: I am quite confused with this paragraph. What are you saying?,What is the
message? Also, some of this seems to be results?
In fact, we are presenting the % of the estimates of genetic distances detected in this
study, comparing with the % of the genetic distances of the study carried out by ZiadiKünzli et al. (2020), in dogfish.
Line 266 and on: see my previous comment about the ABGD-GMYC methods and enrich the
discussion. Beef up the discussion about these methods and include Integrative taxonomy
references.
We revised the text and we add a sentence in the text.
Line 288: This sentence seems to be contradicting itself? Species showing high genetic distance
but also the lowest?
In fact the sentence was a little confused, we checked and reformulated the sentence.
Line 327: A general conclusion is missing here in my opinion? What is the main message and
what will be the next step. Phylogenetic trees, especially of a single marker, do not always
reflect species evolution and phylogenies must be discussed in an integrative framework
that includes all available information (Padial, J.M., Miralles, A., la Riva De, I., Vences,
M., 2010. The integrative future of taxonomy. Front. Zool. 7, 16.)
We revised the text and and improved the ending.
Reviewer 2
line 27: avoid abbreviations in abstract, or else explain what they mean (on first mention)
especially given that they are being used further in the abstract.
We revised the text and correct this sentence.
line 40: genus Squalus rather than Squalus genus
We considered the comment and reformulated the sentence.
line 40-41: Squaliformes, Squalidae - not italics
We checked the format and corrected the typo.
line 41: What is meant by 'three described species'? Is this for a regional perspective or is there
a typo and the statement is global?
We revised the text and correct this sentence. In the literature presented there are 35
valid species, there was a typing error.
line 44: reword the sentence, starting In Brasil, Squalus specimens...
Based on the comment of the other reviewer, we decided to withdraw the sentence
since the genre has a circumglobal distribution and not only in Brazil.
line 49: Is spiny dogfish mentioned for a species specific example S. acanthias (if so include the
scientific name in the text), or is it general and leave it as 'dogfish', as the rest of the
paragraph was general to the genus and then the sentence goes species specific.
In fact, the nomenclature used in the sentence was referring only to one species, but
the information is characteristic of the genus, so we corrected the sentence.
line 63: check style of citations as there are at times spaces between commas and at times no
spaces.
We checked the format that the references were in and standardized.
line 72: There is a study by Naylor that used NADH2 gene for various shark species,
including Squalus; and Vella et al, also used NADH2 together with COI.
We check the jobs and add the references to the sentence.
line 91: How were these species identified? was any morphological ID book or key used?
The samples used in the present work were muscle tissue, for this reason there was no
morphological identification, only molecular identification.
line 96: check units (the 2 must be superscript)
We checked the format and corrected the typo.
line 98: No indication of methodology used, I mean DNA extraction, amplification and
sequencing. There must be at least a few sentences or reference to protocols used.
In fact, this information is very relevant to the research, we introduced a sentence in
the text exemplifying the methodology used.
line 103: do not forget to add accession numbers
The sequences were sent to Genbank, and the access codes were entered in
supplementary table S1.
line 125: re. JC69 model, was this model chosen for some reason or it it the default model in the
software used? I am asking as I think it is important for the reader to know why the model
used here is different from that used in the NJ tree. Maybe even clarify why for the NJ you
used K2P.
The model used In this analysis it was used because it is the default of the program,
we added a sentence explaining the reason for using it.
line 121-142: I believe that this paragraph needs to be reconstructed as it is fairly difficult to
follow the procedure used. Maybe opt for a bullet form system or else number each method
used so that the reader can follow through e.g. 1).... 2)... etc.
We restructured the way the information was presented; in fact it is a little difficult
to understand where an analysis begins and where it ends.
line 162: at times you include a space between value and % and at times the space is missing
We checked the format that the references were in and standardized.
line 166: Include a direct reference to the papers used here as references 58-69 do not appear in
the main text of the manuscript: '204 Squalus sequences obtained from GenBank (Table S1
[references])'. Also may I suggest to include the reference number in Table S1 apart from
the name of the authors.
We have analyzed this comment and added the information containing the references
in the text.
line 174: table 1 repeated words
We checked the format and corrected the typo.
line 179: In table 1 decide whether the decimal is a point or a comma, I would suggest a point to
match the text and the diagonal numbers.
We checked the format that the references were in and standardized.
line 182: change three large clades to three main clades
We checked the sentence and reformulated it based on the comment.
line 188: Figure 1, remove underscore from outgroup
We checked the format and corrected the typo.
line 188: Figure 1, align the boxes of the clades/MOTUs. These MOTUs correspond to numbers
in Table S1? if so link them together by maybe using numbers in Figure 1 matching those
in Table S1 (as currently I am finding table S1 detached from the figure)
We check the image and align the boxes of the MOTUs and the referred numbers
present in Table S1 refer to the number of MOTUs present in the text, we added in
image 1 the number of groups present in each analysis.
line 192: Decide on one format either Generalized Mixed Yule Coalescent or Generalized Mixed
Yule-coalescent throughout the text.
We checked the format that the references were in and standardized.
line 209: Do the numbers of MOTUs mentioned match those in Table S1?
The referred numbers present in Table S1 refer to the number of MOTUs present in
the text, we added in image 1 the number of groups present in each analysis.
line 222: typo 'lineages'
We checked the format and corrected the typo.
line 243: 'The small black dots represent missing haplotypes'. I think this is incorrect, as here
there are no black dots, but rather lines, and if I am understanding well, the lines represent
number of base differences not missing haplotypes, as there is always a line between
haplotypes. So either amend the figure or the caption to clearly indicate what is being
shown.
We revised the text and correct.
line 280: You mention 1% or below 1% difference. Maybe you can include references (either here
or earlier in the introduction) to other shark species where the genetic distance between
species is below 1% e.g. in species within Carcharhinus. You make reference to this gap on
COI throughout the discussion, but there are few references / examples to substantiate this
beyond the genus Squalus. I do recommend that you discuss this in view of other works on
other elasmobranch genera where differences between taxa were too low to delimitate very
closely related elasmobranch species (especially for species that may have diverged
recently)
We revised the text, and were inserted into the introduction and discussion as
suggested by the reviewer.
line 315: S. japonicus, italics
We checked the format and corrected the typo
line 327: Include a conclusion, if needs be, you may reword some statements from the end of the
discussion into a conclusion
We revised the text and and improved the ending.
line 359: references, species names are not italics
We checked the format and corrected the typo
line 363, 366: references, several typos in authors and species names (e.g. missing spaces
between generic name and specific epithet, such as reference number 3. Check all references.
We checked the references and corrected the typo.
Figure S1: Squalus italics
We checked the format and corrected the typo.
Table S1: ID sequence or Sequence ID? (be consistent)
We checked the format that the references were in and standardized.
Table S1: for ABGD, PTP and GMYC, indicate in the caption the meaning of the numbers, that
is the MOTU numbers and how do they relate to Figure 1?
We checked the comment and added to the legend and figure the explanation of the
MOTUs numbers.
Table S1: Include reference numbers
Reference numbers have been included in table S1.

Reviewer 2 Report
Dear authors,
Well done for going through the sequences available for the genus and highlighting the differences between various closely related species.
I have some comments that I would like to share with you to improve some aspects of this work especially in the methodology section to better explain the methods used as they are not written in a user friendly manner. Also, the discussion section needs to be improved with examples from other species, to better corroborate findings and discuss the lack of genetic divergence between taxa outside the genus Squalus. I also recommend the inclusion of a clear conclusion section.
line 27: avoid abbreviations in abstract, or else explain what they mean (on first mention) especially given that they are being used further in the abstract.
line 40: genus Squalus rather than Squalus genus
line 40-41: Squaliformes, Squalidae - not italics
line 41: What is meant by 'three described species'? Is this for a regional perspective or is there a typo and the statement is global?
line 44: reword the sentence, starting In Brasil, Squalus specimens...
line 49: Is spiny dogfish mentioned for a species specific example S. acanthias (if so include the scientific name in the text), or is it general and leave it as 'dogfish', as the rest of the paragraph was general to the genus and then the sentence goes species specific.
line 63: check style of citations as there are at times spaces between commas and at times no spaces.
line 72: There is a study by Naylor that used NADH2 gene for various shark species, including Squalus; and Vella et al, also used NADH2 together with COI.
line 91: How were these species identified? was any morphological ID book or key used?
line 96: check units (the 2 must be superscript)
line 98: No indication of methodology used, I mean DNA extraction, amplification and sequencing. There must be at least a few sentences or reference to protocols used.
line 103: do not forget to add accession numbers
line 125: re. JC69 model, was this model chosen for some reason or it it the default model in the software used? I am asking as I think it is important for the reader to know why the model used here is different from that used in the NJ tree. Maybe even clarify why for the NJ you used K2P.
line 121-142: I believe that this paragraph needs to be reconstructed as it is fairly difficult to follow the procedure used. Maybe opt for a bullet form system or else number each method used so that the reader can follow through e.g. 1).... 2)... etc.
line 162: at times you include a space between value and % and at times the space is missing
line 166: Include a direct reference to the papers used here as references 58-69 do not appear in the main text of the manuscript: '204 Squalus sequences obtained from GenBank (Table S1 [references])'. Also may I suggest to include the reference number in Table S1 apart from the name of the authors.
line 174: table 1 repeated words
line 179: In table 1 decide whether the decimal is a point or a comma, I would suggest a point to match the text and the diagonal numbers.
line 182: change three large clades to three main clades
line 188: Figure 1, remove underscore from outgroup
line 188: Figure 1, align the boxes of the clades/MOTUs. These MOTUs correspond to numbers in Table S1? if so link them together by maybe using numbers in Figure 1 matching those in Table S1 (as currently I am finding table S1 detached from the figure)
line 192: Decide on one format either Generalized Mixed Yule Coalescent or Generalized Mixed Yule-coalescent throughout the text
line 209: Do the numbers of MOTUs mentioned match those in Table S1?
line 222: typo 'lineages'
line 243: 'The small black dots represent missing haplotypes'. I think this is incorrect, as here there are no black dots, but rather lines, and if I am understanding well, the lines represent number of base differences not missing haplotypes, as there is always a line between haplotypes. So either amend the figure or the caption to clearly indicate what is being shown.
line 280: You mention 1% or below 1% difference. Maybe you can include references (either here or earlier in the introduction) to other shark species where the genetic distance between species is below 1% e.g. in species within Carcharhinus. You make reference to this gap on COI throughout the discussion, but there are few references / examples to substantiate this beyond the genus Squalus. I do recommend that you discuss this in view of other works on other elasmobranch genera where differences between taxa were too low to delimitate very closely related elasmobranch species (especially for species that may have diverged recently)
line 315: S. japonicus, italics
line 327: Include a conclusion, if needs be, you may reword some statements from the end of the discussion into a conclusion
line 359: references, species names are not italics
line 363, 366: references, several typos in authors and species names (e.g. missing spaces between generic name and specific epithet, such as reference number 3. Check all references.
Figure S1: Squalus italics
Table S1: ID sequence or Sequence ID? (be consistent)
Table S1: for ABGD, PTP and GMYC, indicate in the caption the meaning of the numbers, that is the MOTU numbers and how do they relate to Figure 1?
Table S1: Include reference numbers
Author Response

(The authors gave the same response as above.)
